# Impact of Heat Stress on Milk Yield, Milk Fat-to-Protein Ratio, and Conception Rate in Thai–Holstein Dairy Cattle: A Phenotypic and Genetic Perspective

**DOI:** 10.3390/ani14203026

**Published:** 2024-10-19

**Authors:** Wuttigrai Boonkum, Watcharapong Teawyoneyong, Vibuntita Chankitisakul, Monchai Duangjinda, Sayan Buaban

**Affiliations:** 1Department of Animal Science, Faculty of Agriculture, Khon Kean University, Khon Kean 40002, Thailand; majehow@hotmail.com (W.T.); vibuch@kku.ac.th (V.C.); monchai@kku.ac.th (M.D.); 2Network Center for Animal Breeding and Omics Research, Khon Kaen University, Khon Kaen 40002, Thailand; 3Bureau of Animal Husbandry and Genetic Improvement, Department of Livestock Development, Pathum Thani 12000, Thailand; buaban_ai@hotmail.com

**Keywords:** heat tolerance, genetic parameter, multiple traits, Thai dairy cattle

## Abstract

Environmental indices are commonly used for detecting heat stress in dairy cattle; however, most studies have only focused on the productivity traits of dairy cattle in response to heat stress and have not considered their health and reproductive characteristics simultaneously. In this study, we aimed to determine the effects of heat stress on the production and reproduction performances of Thai–Holstein dairy cattle and the impact of the genetics of dairy cattle on their heat tolerance. We observed that heat stress significantly reduced milk yield and negatively affected the milk fat-to-protein ratio. Additionally, conception rates declined under heat stress, highlighting the challenges of finding an effective genetic approach for hot and humid regions, including Thailand. Genetic analysis revealed differences in heat tolerance among cows, indicating that the genetic improvement approach used in this study is suitable for planning future genetic improvements in dairy cattle.

## 1. Introduction

Heat stress threatens dairy farming systems worldwide, particularly in tropical regions, including Thailand. In these areas, the challenges of dairy farming are heightened by consistently high temperatures and humidity levels throughout the year [1,2]. Dairy farming contributes 12.6% of Thailand’s gross agricultural output, driven by increasing demand for animal-based food domestically and internationally [3,4]. Most farms are small-scale, providing income stability to rural communities, especially in the Central and Northeastern regions, which produce 70% of the country’s milk [5]. The dairy industry meets domestic demand, strengthens food security, and promotes self-sufficiency. Enhancing productivity through improved breeding is crucial for ensuring future economic resilience [6]. Various strategies have been implemented to mitigate heat stress, including environmental modifications such as the use of thermal stress relief equipment [7,8,9], adjustments in feeding and nutrition [10,11,12], and the development of crossbreeding systems between *Bos taurus* and *Bos indicus* [5,13]. However, these methods have certain limitations, such as high initial investment costs, short-term efficacy, and the need for ongoing adjustments to match changing environmental conditions. Therefore, alternative methods offering sustainable solutions are required.

Genetic selection methods such as genetic evaluation [14,15,16,17], marker-assisted selection [18,19], Genome-Wide Association Studies [20,21,22,23], and genomic selection [2,24,25] are widely used in countries like Australia, Kenya, Mexico, the United States, Tunisia, and Thailand to study the effects of heat stress on dairy cattle phenotypes and genetics. In particular, genetic evaluation is often the preferred method due to its suitability for large datasets and lower cost compared to other approaches. This method combines genetic improvement of economically important traits with enhanced adaptation to extreme climatic conditions such as heat stress [26,27,28,29]. It relies on genetic evaluation using the temperature and humidity index (THI) to assess genetic parameters and estimate breeding values under various environmental conditions [28,30,31,32]. Previous studies have demonstrated the effectiveness of genetic selection methods in animal breeding [33,34,35]. These methods are applicable across diverse populations, ranging from large to small, and tend to have lower operational costs than those associated with other interventions [33,36,37]. However, research on genetic selection using THI has primarily focused on purebred dairy cows, with few studies on crossbred counterparts [24,38,39]. Moreover, although their productivity traits have been studied, there is a lack of scientific data regarding their health and reproductive traits, which are crucial. Understanding these traits is essential, as they can directly affect productivity, considering the interrelationships between production, health, and reproduction [40,41].

Focusing on effective genetic selection that encompasses yield, health, and reproductive traits is crucial while developing genetic resilience to heat stress to achieve sustainability in dairy production systems. In Thailand, crossbreeding between *Bos taurus* breeds (such as Holstein, Jersey, Brown Swiss, and Red Dane) and *Bos indicus* breeds (such as Sahiwal, Red Sindhi, Brahman, and Thai Native) is widely practiced. However, the majority of crossbred dairy cattle (>95%) are the result of crosses between Holstein and either Sahiwal or Thai Native breeds [5,28]. This is necessary in tropical climates, where these traits are highly valued. Milk yield (MY) is the most critical economic trait for improving farmers’ income and, therefore, a fundamental target in genetic improvement programs. Additionally, a lack of energy in the body leads to increased lipolysis, which boosts fat synthesis in the mammary glands. Simultaneously, an insufficient intake of digestible carbohydrates to meet the body’s demands reduces protein synthesis by gut bacteria. This imbalance is reflected in a change in the milk fat-to-protein ratio (FPR), an indirect indicator of the cow’s health and reproductive status, ultimately leading to decreased fertility and broader health issues [42,43]. Regarding reproductive traits, the percentage of successful pregnancies in a cow after insemination or mating, called conception rate (CR) along the stage of lactation, is a key indicator of reproductive efficiency in dairy herds. High CRs indicate the regular estrous cycles and successful pregnancies necessary for maintaining a productive herd [44]. However, low CR is a significant reason for culling cows from herds. Selecting heat tolerance in dairy cattle based on multiple traits, such as optimal MY, milk FPR, and CR, is a complex challenge affecting farmers’ profitability. However, studies on heat stress in dairy cows often focus on individual traits or specific groups of productivity or reproductive characteristics, which may not fully align with the multifaceted needs of farmers seeking simultaneous genetic improvements across various attributes. We hypothesized that heat stress affects both yield and reproductive traits; however, the timing and severity of these effects remain unclear. Therefore, in this study, we aimed to investigate this issue using a genetic approach to elucidate the genetic parameters affecting production and reproduction performances, such as MY, milk FPR, and CR under heat stress conditions in dairy cattle, and to select for heat tolerance in crossbred Thai–Holstein dairy cattle.

## 2. Materials and Methods

### 2.1. Data for Analysis

The Institutional Animal Care and Use Committee of Khon Kaen University reviewed and approved this study in accordance with the Ethics of Animal Experimentation Guidelines set by the National Research Council of Thailand (No. IACUC-KKU-120/64; 30 November 2021). A total of 168,124 records for MY and milk FPR, along with 21,278 records for CR (conception status determined by ultrasound 30 days after breeding, where 2 = success and 1 = failure), were collected from 21,278 first-lactation crossbred Thai–Holstein cows. These records, provided by the Bureau of Biotechnology in Livestock Production, Department of Livestock Development, Thailand, were gathered between 1999 and 2017. The cows were grouped based on the percentage of Holstein genetics (breed group; BG), with BG1, BG2, and BG3 representing <87.5, 87.5–93.6, and >93.7% of Holstein genetics, respectively. Age at first service was divided into seven classes at 3-month intervals, with the first class being <25 months and last class >39 months. A summary of these data is presented in Table 1.

Climate data were obtained from the Thai Meteorological Department based on the postal code closest to each dairy farm. These weather data were linked to each trait (MY, milk FPR, and CR) and used to calculate THI. The weather data included daily air temperature in degrees Celsius (Temp, °C) and relative humidity (RH, %) recorded automatically every 3 h by a digital device and were subsequently used to calculate the THI based on the formula provided by the National Oceanic and Atmospheric Administration [45]. The THI function was as follows.
THI = (1.8 × Temp + 32) − (0.55 − 0.0055 × RH) × (1.8 × Temp − 26)

The average daily THI estimates, beginning 3.0 d before the collection of MY and milk FPR records and 30 d before the collection of CR records, were used to determine the threshold point of heat stress and estimate variance components and genetic parameters, following the method suggested by Bohmanova et al. [46].

### 2.2. Analysis of the THI Threshold Point of Heat Stress

Various thresholds, specifically THI72–THI84, were tested in the multiple-trait threshold-linear random regression model. The first step involved analyzing the simple linear regression between THI values and MY, milk FPR, and CR. If the initial regression coefficient (slope) at any THI value was negative, it was considered the critical value for heat stress. Additionally, the THI values used in this model were set at critical points. The best model was determined based on the lowest negative twice the Log-Likelihood (−2logL) and lowest Akaike’s information criterion (AIC), identified as the “threshold point of heat stress”.

### 2.3. Genetic Model and Variance Component Estimation

The three traits (MY, milk FPR, and CR) were analyzed using the multiple-trait threshold-linear random regression model to estimate variance components, genetic parameters, and the rates of decline for each trait. The model was as follows.
$$\left[\begin{array}{c} y\\ l \end{array}\right]\;=\;\left[\begin{array}{c} X\beta \;+\;W_{1}h\;+\;Z_{1}a\;+\;Z_{2}p\;+\;e\\ X\beta \;+\;W_{1}h{\;+\;W}_{2}s\;+\;Z_{1}a\;+\;Z_{2}p\;+\;e \end{array}\right],$$
where y represents the vectors of observations for MY and milk FPR records as continuous traits; l represents the vectors of unobserved liabilities for CR based on a binary outcome (no conception or conception); β represents the vectors of fixed effects, including herd × test-month × test-year for MY and milk FPR, and herd × year × season for CR, age at first calving, months in milk, and the fixed regression coefficient for the rate of decline in traits per THI level, separated based on BG; h represents the vector of random herd effects; s represents the vector of random service sire effects used only for CR; a represents the vector of random animal effects with and without the THI; p represents the vector of random permanent environmental effects for cows with and without THI; and e represents the vector of random residual effects. The incidence matrices $X,\;W_{1},\;W_{2},\;Z_{1},\;and\;Z_{2}$ correspond to β, h, s, a, and p, respectively.

The (co)variance structures were defined as follows.
$$Var\left[\begin{array}{c} h\\ a\\ \begin{array}{c} p\\ e \end{array} \end{array}\right]=\left[\begin{array}{ccc} {I⊗H}_{0} & 0 & 0\\ 0 & {A⊗G}_{0} & 0\\ \begin{array}{c} 0\\ 0 \end{array} & \begin{array}{c} 0\\ 0 \end{array} & \begin{array}{c} {I⊗P}_{0}\\ 0 \end{array} \end{array}\begin{array}{c} 0\\ 0\\ \begin{array}{c} 0\\ {I⊗R}_{0} \end{array} \end{array}\right]\;(for\;MY\;and\;milk\;FPR\;traits)$$
and
$$\mathrm{Var}\left[\begin{array}{c} h\\ s\\ \begin{array}{c} a\\ p\\ e \end{array} \end{array}\right]=\left[\begin{array}{ccc} {I⊗H}_{0} & 0 & 0\\ 0 & {I⊗S}_{0} & 0\\ \begin{array}{c} 0\\ 0\\ 0 \end{array} & \begin{array}{c} 0\\ 0\\ 0 \end{array} & \begin{array}{c} {A⊗G}_{0}\\ 0\\ 0 \end{array} \end{array}\begin{array}{c} 0\\ 0\\ \begin{array}{c} 0\\ {I⊗P}_{0}\\ 0 \end{array} \end{array}\begin{array}{c} 0\\ 0\\ \begin{array}{c} 0\\ 0\\ {I⊗R}_{0} \end{array} \end{array}\right]\;(for\;CR\;trait),$$
where $H_{0}$ represents the herd variance matrix; $S_{0}$ is the service sire variance matrix for CR; $G_{0}$ and $P_{0}$ are 6 × 6 (co)variance matrices for general and heat stress additive and permanent environmental effects, respectively; $R_{0}$ is the diagonal matrix of the residual variances for each trait; A is the numerator relationship matrix; and I is the identity matrix. Variance components, genetic parameters (heritability and repeatability), and estimated breeding values were calculated using a Bayesian approach via Gibbs sampling. Computations were performed using the THRGIBBS1F90 program [47]. A total of 500,000 iterations were conducted, with the first 50,000 samples discarded as burn-in and every 10th sample retained subsequently. Post-Gibbs analysis was performed using the POSTGIBBSF90 program [47] to obtain posterior distribution statistics to verify parameter estimates.

### 2.4. Genetic Parameter Estimation

The heritability equations (*h*^2^) under heat stress conditions for MY, milk FPR, and CR under heat stress applied by Ravagnolo and Misztal [48] are as follows.
$$h^{2}=\frac{\sigma_{a}^{2}+\sigma_{v}^{2}+{2\sigma }_{av}}{\sigma_{a}^{2}+\sigma_{v}^{2}+{2\sigma }_{av}+\sigma_{p}^{2}+\sigma_{q}^{2}+{2\sigma }_{pq}+\sigma_{h}^{2}+\sigma_{e}^{2}}\;(for\;MY\;and\;milk\;FPR\;traits)$$
and
$$h^{2}=\frac{\sigma_{a}^{2}+\sigma_{v}^{2}+\sigma_{av}}{\sigma_{a}^{2}+\sigma_{v}^{2}+{2\sigma }_{av}+\sigma_{p}^{2}+\sigma_{q}^{2}+{2\sigma }_{pq}+\sigma_{h}^{2}+\sigma_{s}^{2}+\sigma_{e}^{2}}\;(for\;CR\;trait),$$
where $\sigma_{a}^{2}$ represents additive genetic variance; $\sigma_{v}^{2}$ denotes additive genetic variance under heat stress; $\sigma_{av}$ is the covariance between additive genetic variance and heat stress variance; $\sigma_{p}^{2}$ indicates permanent environmental variance; $\sigma_{q}^{2}$ refers to permanent environmental variance under heat stress; $\sigma_{pq}$ is the covariance between permanent environmental variance and heat stress variance; $\sigma_{h}^{2}$ represents herd variance; $\sigma_{s}^{2}$ is the service sire variance; and $\sigma_{e}^{2}$ denotes residual variance.

Genetic correlations ($r_{g}$) and permanent environmental effect correlations ($r_{p}$) between the studied traits and heat stress were analyzed using the equation provided by Ravagnolo and Misztal [48] as follows.
$$r_{g}=\frac{\sigma_{av}}{\sqrt{\sigma_{a}^{2}\times \sigma_{v}^{2}}}$$
and
$$r_{p}=\frac{\sigma_{pq}}{\sqrt{\sigma_{p}^{2}\times \sigma_{q}^{2}}}$$

## 3. Results

### 3.1. Descriptive Statistics

Table 1 presents the data used in this study. The data from BG1, BG2, and BG3 showed differences in MY, milk FPR, and CR. Regarding MY, BG3 had the highest mean yield at 13.78 kg, followed by those in BG2 and BG1 at 12.85 and 12.74 kg, respectively. The standard deviation was the largest in BG3 (4.15), indicating a greater variation in MY within this group than in the other groups. The maximum yield was the highest in BG3 at 40.20 kg, whereas the minimum yield was consistent at 7.00 kg across all groups. Regarding the FPR, BG1 had the highest mean ratio at 1.19, slightly above that of BG2 at 1.17 and BG3 at 1.16. The standard deviation was largest in BG3 (0.31), and BG1 had the highest maximum ratio at 4.37, suggesting a greater variability in fat–protein content in BG1 than in the other groups. Lastly, the CR was the highest in BG1 at 34.02%, followed by those in BG2 at 33.52% and BG3 at 32.73%, which indicates that BG1 had slightly better reproductive success than that exhibited by BG2 and BG3, although the differences were relatively small.

Environmental conditions were recorded with an average RH of 73.55%, ranging from 47.48 to 90.56%, and an average temperature of 27.78 °C, which ranged from 19.92 to 33.80 °C. The THI had an average value of 78.50, with a standard deviation of 4.22, and minimum and maximum THI values of 66.48 and 84.52, respectively.

### 3.2. Determination of Appropriate THI Threshold

The THI threshold in this study was determined based on regression coefficients and −2 logL and AIC values between THI levels and MY, milk FPR, and CR traits (Table 2). As THI levels increased, a noticeable decline in all three parameters was observed. Specifically, at a THI of 76, negative regression coefficients were observed for MY (−0.284), milk FPR (−0.094), and CR (−0.089), indicating the onset of heat stress effects. This trend continued, with the most substantial negative impacts occurring at THI84, where MY, milk FPR, and CR had slopes of −0.421, −0.250, and −0.315, respectively. Additionally, the −2logL and AIC values were lowest at THI76. The results indicate that heat stress began to have an additive detrimental effect on MY, milk FPR, and CR traits when THI levels exceeded 76. Expanding THI76 in terms of air temperature and RH, it was found that air temperatures ranging from 25.6 to 26.9 °C and RH between 60.0 and 68.6% marked the onset of heat stress in the Thai dairy cattle population.

### 3.3. Estimated Variance Components and Heritability Values

The estimated variance components and heritability values under heat stress for MY, milk FPR, and CR at THI76 and THI80 are presented in Table 3. The additive genetic variance for MY was slightly reduced from 10.346 at THI76 to 10.201 at THI80, while the heritability for MY remained relatively stable, with values of 0.380 at THI76 and 0.377 at THI80. For milk FPR, the additive genetic variance decreased from 0.662 at THI76 to 0.622 at THI80, but heritability stayed the same at 0.293. The CR trait showed a more noticeable reduction in both variance and heritability, with heritability dropping from 0.032 at THI76 to 0.026 at THI80. These results suggest that heat stress, especially at severe levels (THI80), has a more pronounced effect on CR than on M or milk FPR.

### 3.4. Genetic Correlations and Permanent Environmental Effects

Genetic correlations and permanent environmental effects between the studied traits and heat tolerance are shown in Table 4. The genetic correlations of MY with milk FPR and CR were −0.24 and −0.53, respectively, which indicates that a higher MY is genetically associated with a lower milk FPR and CR. The genetic correlation between MY and heat tolerance was −0.26, suggesting that when dairy cows have a high genetic potential for MY, their genetic tolerance to heat stress decreases. The genetic correlation between FPR and heat tolerance was −0.48, showing a moderately negative relationship and indicating that higher FPR is genetically linked to lower heat stress tolerance. CR exhibited a negative genetic correlation with heat stress effects at −0.49, indicating that a high CR is genetically associated with reduced heat stress tolerance.

Regarding permanent environmental effects, MY and milk FPR had a slight negative correlation of −0.04, whereas MY and CR exhibited a stronger negative correlation of −0.25. The permanent environmental correlation between milk FPR and CR was 0.38, indicating a moderately positive relationship. All three traits—MY, milk FPR, and CR—showed strong negative permanent environmental correlations with heat stress, with values of −0.68, −0.67, and −0.71, respectively.

### 3.5. Rates of Decline in MY, Milk FPR, and CR

The decrease in values for each trait because of heat stress at the THI levels of 76 and 80 was associated with the BG of cows based on Holstein genetics (Figure 1). The results indicate that an increase in THI from 76 to 80 generally exacerbates the decline in MY, milk FPR, and CR, although the magnitude and direction of these changes varied among the different BGs. For BG1, the decline in MY was 0.046 kg at THI76 and 0.066 kg at THI80, representing an approximately 43.5% increase in the rate of decline when THI increased from 76 to 80. For BG2 and BG3, the declines in MY were 0.086 and 0.116 kg at THI76 and 0.073 and 0.218 kg at THI80, reflecting a reduction of approximately 15.1% and an increase of 87.9%, respectively, in the decline rate. Moreover, the declines in FPR were 0.015, 0.045, and 0.064 at THI76 and 0.021, 0.021, and 0.095 at THI80, indicating 40, 53.3, and 48.4% increases in the decline rate for BG1, BG2, and BG3, respectively. For the CR trait, the declines for BG1, BG2, and BG3 were 0.027, 0.102, and 0.120% at THI76 and 0.064, 0.050, and 0.179% at THI80, reflecting an increase of approximately 137, 51, and 49.2% from THI76 to THI80, respectively.

## 4. Discussion

Heat stress negatively impacts dairy cattle performance. There is no single strategy to fully mitigate the adverse effects of heat stress in dairy farming; however, genetic improvement methods may offer better and relatively more sustainable solutions.

Based on regression coefficients and −2LogL and AIC values, in this study, we found that the starting point of heat stress (THI threshold) in the Thai–Holstein population was at THI76, which is higher than that reported in previous studies. Regarding MY and FPR traits, Brugemann et al. [49] reported a substantial decline in daily MY at THI above 60, whereas Zimbelman et al. [50] evaluated the effects of THI values on changes in the MY of dairy cows in the southern United States, declaring a critical THI threshold of 68. Heinicke et al. [51] reported that the heat load threshold determined using piecewise models is THI67 for lactating dairy cows in a temperate climate in Germany. In addition, compared with the results reported by Boonkum and Duangjinda [28], who studied the same dairy population in 2015, it was found that the current dairy population had a higher critical value for heat stress than that exhibited by past dairy populations. Regarding CR, Schüller et al. [52] reported that the THI threshold for the impact of heat stress on CR was 73, whereas Morton et al. [53] reported a decrease in CR with a THI threshold of 72. However, some reported THI thresholds were higher than those reported in this study, such as that of Michael et al. [54], who reported four THI functions in dairy farms in Malaysia and found that the THI threshold ranged from 73 to 79. Moreover, Pramod et al. [55] reported that the THI threshold in the humid tropical climate of India is 72–79. Bohmanova et al. [32] reported that the impact of heat stress on dairy cows in semi-arid and hot and humid climates in the United States have critical values ranging from 68 to 83, depending on the THI function used and the area, with higher critical values in hot and humid climates than in semi-arid climates.

The THI threshold for heat stress in this study was higher than that reported in many previous studies, suggesting that Thai dairy cows have better heat tolerance ability than that observed in dairy cows from many other regions. The main reasons for this are presented below. Thailand has implemented a dairy breeding program focusing on crossbreeding Holstein cows with local Thai breeds. This combination aids in combining desirable traits such as the high MY of Holstein cattle and heat tolerance of local breeds [5,28]. Additionally, selecting heat tolerant breeds that are well-adapted to the environment in Thailand is essential for developing this hybrid. Consequently, Thai–Holstein crossbred dairy cows show strong physiological adaptations, including appropriate respiratory and sweating rates, efficient cooling mechanisms to reduce body temperature, and effective water balance control (packed cell volume), enabling them to thrive in hot climates [56,57,58]. In conclusion, using THI data from weather stations can help study the effects of heat stress on the phenotype and genetics of the Thai–Holstein population and reduce costs for farmers by eliminating the need to collect weather data; however, relatively more accurate evaluations could be achieved by incorporating rumen temperature measurements [12] or surface temperature measurements [36].

Variance components and heritability values estimated for MY, milk FPR, and CR under THI76 and THI80 provide key insights into genetic and environmental factors impacting these traits under heat stress conditions. Heritability estimates (*h*^2^) are crucial in elucidating the extent to which these traits are genetically determined and their potential for improvement through selective breeding. The heritability of MY (0.380 ± 0.032 and 0.377 ± 0.030 for THI76 and THI80) indicates that moderate variations in MY can be attributed to genetic factors, suggesting that selective breeding can improve this trait even under heat stress, which is supported by high additive genetic variance ($\sigma_{a}^{2}$) for MY, indicating that genetic selection can considerably impact this trait. However, the heritability of milk FPR (0.293 ± 0.021 and 0.293 ± 0.020 for THI76 and THI80) is slightly lower but moderate, implying that while genetic selection could enhance this trait, it may require relatively more careful management to achieve substantial improvements, particularly in stressful environmental conditions. The additive genetic variance under heat stress ($\sigma_{v}^{2}$) for milk FPR is relatively small, highlighting the challenges presented by heat stress in maintaining the desired milk FPR. The CR exhibits a much lower heritability (0.032 ± 0.001 and 0.026 ± 0.001 for THI76 and THI80), indicating that genetic factors play a minimal role in the variation of this trait. This low heritability suggests that CR is relatively more impacted by environmental factors rather than genetics, making it challenging to improve CR through traditional breeding programs. In addition, the heritability decreases from 0.032 at THI 76 to 0.026 at THI 80, reflecting a more pronounced impact of severe heat stress on reproductive traits. The substantial service sire variance ($\sigma_{s}^{2}$) observed for CR further emphasizes the impact of external factors, such as management practices and environmental conditions, on reproductive performance.

The negative covariance between additive genetic variance and heat stress variance ($\sigma_{av}$) across all traits underscored the antagonistic relationship between heat stress and genetic potential for these traits. This negative covariance suggests that genetic improvements in these traits may be compromised under heat stress, making it crucial to consider heat tolerance in breeding programs. Additionally, relatively high residual variance ($\sigma_{e}^{2}$) for MY and moderate herd variance ($\sigma_{h}^{2}$) for all traits indicate that non-genetic factors, including management practices and environmental conditions, particularly impact these traits. These findings highlight the importance of optimizing environmental and management conditions, along with genetic selection, to maximize the performance of these traits under heat stress. In conclusion, although genetic selection can moderately improve MY and milk FPR under heat stress, CR remains largely impacted by environmental factors. Breeding strategies should integrate heat tolerance to enhance the overall performance of dairy cattle, particularly under variable and stressful climatic conditions.

While Thai cattle, renowned for their resilience in tropical climates, likely possess specific genetic adaptations that contribute to heat tolerance, the present study revealed negative correlations between milk production (MY and milk FPR) and heat tolerance, as well as reproductive performance (CR) and heat tolerance (Table 4). These negative correlations can be explained by the complex interplay of physiological and metabolic responses to heat stress. Heat stress significantly impacts a cow’s appetite, directly limiting the energy and nutrients available for milk production and affecting milk composition [59,60]. The cow’s energy is then prioritized for survival over fat synthesis, leading to a lower milk FPR [61]. Furthermore, heat stress disrupts the production and mobilization of fatty acids, crucial for milk fat synthesis [14,62], further contributing to the decline in milk FPR. These combined factors demonstrate how heat stress negatively affects a cow’s milk fat-to-protein ratio, highlighting the importance of managing heat stress in dairy farming operations.

Similarly, heat stress significantly impacts a cow’s reproductive performance, particularly affecting conception rate and embryo viability. One key mechanism is the damage to oocyte quality during folliculogenesis, leading to lower fertilization rates and embryo viability [63]. Elevated cortisol levels and reduced progesterone levels in heat-stressed cows are also associated with decreased conception rates and impaired embryo development [64]. The first 16 days post-fertilization are a critical window for embryo development; heat stress can stunt embryo growth and reduce the number of transferable embryos during this crucial period [64]. Moreover, we demonstrate that the correlation between CR and milk production is also negative (−0.53), as evidenced by various studies [65,66]. Cows with higher milk production often have decreased reproductive performance because nutrients are prioritized for milk production over reproductive development [44].

These correlations underscore the importance of considering these factors in breeding programs to ensure a balance between production traits and heat tolerance. While genetics play a crucial role, it is equally important to acknowledge that environmental factors significantly influence heat tolerance. Management practices, feeding strategies, and the availability of shade or cooling systems can all influence a dairy cow’s ability to cope with heat stress. Furthermore, permanent environmental effects highlight the effects of non-genetic factors on these traits. For instance, low-to-moderate negative environmental correlations suggest that environmental factors affecting these traits are relatively consistent across the different traits, reinforcing the need to manage these factors alongside genetic selection to optimize overall performance. Ultimately, understanding the complex interplay between genotype, phenotype, and environmental factors is essential for developing successful breeding strategies that improve heat tolerance in Thai dairy cattle. By carefully considering both genetic and environmental factors, breeders can optimize production while promoting the health and well-being of cows in hot climates.

The rates of decline in the studied traits indicate that crossbred dairy cows with a high proportion of temperate blood have a reduced ability to withstand heat stress compared with that exhibited by crossbred dairy cows with a high proportion of tropical blood. As THI increases, the rates of decline in these characteristics also increases. These differences can be attributed to genetic adaptation, physiology, and evolutionary history. Owing to these factors, dairy cows from cold regions are relatively less heat tolerant than those from tropical regions [67] and have evolved traits to conserve heat, such as thicker hair coats and higher metabolic rates, advantageous in cold weather but detrimental in hot conditions [57,68,69]. In contrast, cows in tropical regions have developed characteristics such as light hair coats, relatively more efficient sweat glands, and behaviors that reduce heat stress, which help them dissipate heat more effectively [58,70]. These adaptations result from generations of natural selection in their respective environments; tropical cows face constant heat stress and have evolved mechanisms to cope, whereas temperate cows do not. Consequently, the physiological and genetic makeup of cows in temperate regions makes them relatively less equipped to handle high temperatures than their tropical counterparts. However, what is concerning is that the rate of decline in all three productivity traits was much higher than that reported in past studies [2,52].

## 5. Conclusions

Addressing heat stress is a primary challenge in modern dairy farming under hot climatic conditions. Our findings indicate that heat stress reduces MY, milk FPR, and CR, considering THI values from 76 onwards. The effects were relatively more severe in terms of both phenotypic and genetic expression at higher THI values and with a higher percentage of Holstein genetics. Therefore, considering the expected increase in the frequency and severity of heat events, additional strategies, including selection and breeding for thermotolerance, are required.

## Figures and Tables

**Figure 1 animals-14-03026-f001:**
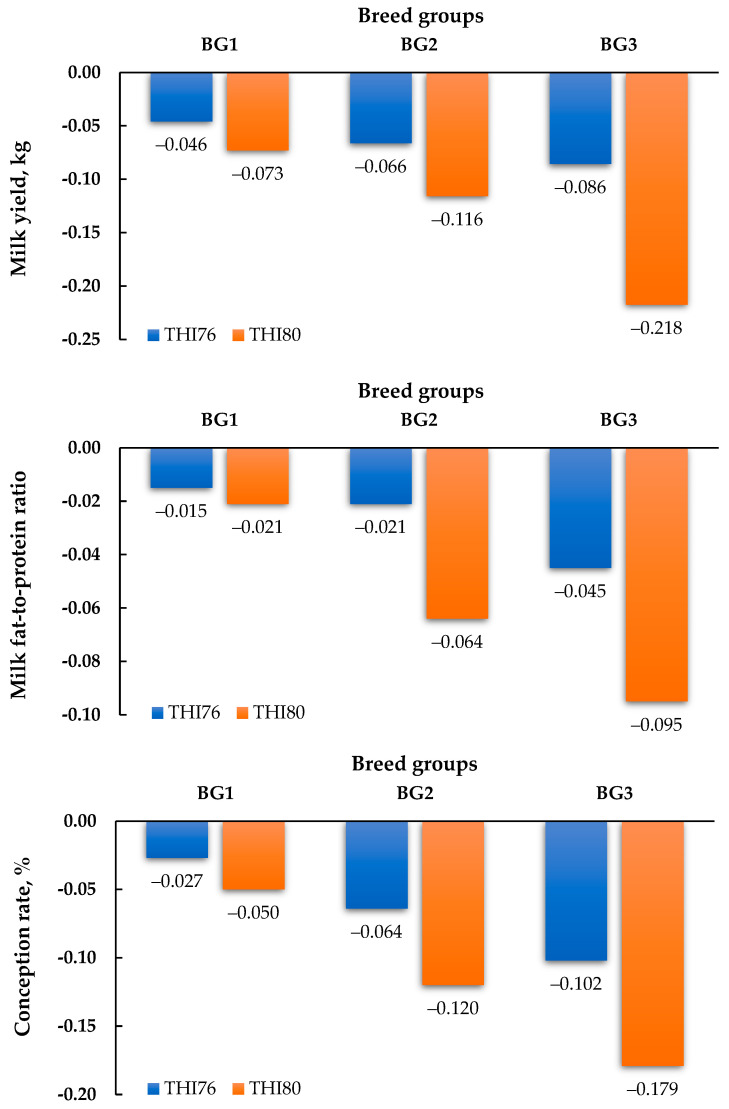
Rates of decline in milk yield, milk fat-to-protein ratio, and conception rate at the temperature and humidity indexes (THIs) of 76 (blue color bar) and 80 (orange color bar) separated based on breed groups (BG1, BG2, and BG3 represent <87.5, 87.5–93.6, and >93.7% Holstein genetics, respectively).

**Table 1 animals-14-03026-t001:** Data structure for analysis.

Categories	Number	Mean	SD	Minimum	Maximum
Animals with data records, n	21,278	-	-	-	-
Animals with pedigrees, n	46,198	-	-	-	-
Herd, n	63	-	-	-	-
Milk yield, kg	168,124	13.49	4.10	7.00	45.00
BG1	11,278	12.74	3.93	7.00	36.50
BG2	40,738	12.85	3.91	7.00	37.10
BG3	116,108	13.78	4.15	7.00	40.20
Milk fat-to-protein ratio	168,124	1.16	0.30	0.23	4.37
BG1	11,278	1.19	0.30	0.24	4.37
BG2	40,738	1.17	0.28	0.26	3.26
BG3	116,108	1.16	0.31	0.23	3.75
Conception rate, %	21,278	33.45	-	-	-
BG1	1410	34.02	-	-	-
BG2	5092	33.52	-	-	-
BG3	14,776	32.73	-	-	-
Relative humidity, %	-	73.55	3.24	47.48	90.56
Air temperature, °C	-	27.78	2.11	19.92	33.80
Temperature and humidity index	-	78.50	4.22	66.48	84.52

Abbreviations: SD, standard deviation; BG, breeding group.

**Table 2 animals-14-03026-t002:** Regression coefficient (slope), negative twice the Log-Likelihood (−2logL), and Akaike’s information criterion (AIC) values in various temperature and humidity index levels to determine the threshold point of heat stress for milk yield, milk fat-to-protein ratio, and conception rate.

THI Levels	Slope			−2logL			AIC		
	MY	Milk FPR	CR	MY	Milk FPR	CR	MY	Milk FPR	CR
THI72	0.215	0.012	0.040	80	66	145	80	66	145
THI73	0.226	0.008	0.022	74	52	122	74	52	122
THI74	0.134	0.004	0.005	62	43	88	62	43	88
THI75	0.055	0.001	0.000	30	20	53	30	20	53
THI76	−0.284	−0.094	−0.089	0	0	0	0	0	0
THI77	−0.305	−0.122	−0.129	22	15	43	22	15	43
THI78	−0.322	−0.128	−0.144	40	38	67	40	38	67
THI79	−0.329	−0.135	−0.167	65	59	98	65	59	98
THI80	−0.350	−0.159	−0.195	89	72	179	89	72	179
THI81	−0.365	−0.172	−0.244	129	99	266	129	99	266
THI82	−0.382	−0.210	−0.282	237	182	357	237	182	357
THI83	−0.390	−0.239	−0.299	338	288	598	338	288	598
THI84	−0.421	−0.250	−0.315	653	436	744	653	436	744

**Table 3 animals-14-03026-t003:** Variance components and heritability values under heat stress of milk yield, milk fat-to-protein ratio, and conception rate at a temperature and humidity index of 76 and 80 (THI76, and THI80).

Variance Components and Heritability Values	THI76			THI80		
	MY	Milk FPR	CR	MY	Milk FPR	CR
$\sigma_{a}^{2}$	10.346	0.662	0.031	10.201	0.622	0.025
$\sigma_{v}^{2}$	0.312	0.015	0.005	0.334	0.020	0.012
$\sigma_{av}$	−0.463	−0.048	−0.005	−0.477	−0.064	−0.008
$\sigma_{p}^{2}$	9.085	1.881	0.001	8.988	1.866	0.000
$\sigma_{q}^{2}$	0.936	0.146	0.008	1.242	0.252	0.019
$\sigma_{pq}$	−0.973	−0.353	−0.002	−1.211	−0.488	−0.010
$\sigma_{h}^{2}$	4.070	0.010	0.760	4.090	0.011	0.770
$\sigma_{s}^{2}$	-	-	12.190	-	-	12.220
$\sigma_{e}^{2}$	3.760	0.070	0.010	3.920	0.090	0.010
$h^{2}\;$± SE	0.380 ± 0.032	0.293 ± 0.021	0.032 ± 0.001	0.377 ± 0.030	0.293 ± 0.020	0.026 ± 0.001

$\sigma_{a}^{2}$ = additive genetic variance; $\sigma_{v}^{2}$ = additive genetic variance under heat stress; $\sigma_{av}$ = covariance between additive genetic variance and heat stress variance; $\sigma_{p}^{2}$ = permanent environmental variance; $\sigma_{q}^{2}$ = permanent environmental variance under heat stress; $\sigma_{pq}$ = covariance between permanent environmental variance and heat stress variance; $\sigma_{h}^{2}$ = herd variance; $\sigma_{s}^{2}$ = service sire variance; $\sigma_{e}^{2}$ = residual variance; $h^{2}$ = heritability; SE = standard error.

**Table 4 animals-14-03026-t004:** Genetic correlations (above diagonal) and permanent environmental effects (below diagonal) between milk yield, milk fat-to-protein ratio, conception rate, and heat tolerance at a temperature and humidity index of 76.

Traits	MY	Milk FPR	CR	Heat Tolerance
MY	-	−0.24	−0.53	−0.26
Milk FPR	−0.04	-	0.48	−0.48
CR	−0.25	0.38	-	−0.49
Heat tolerance	−0.68	−0.67	−0.71	-

## Data Availability

The data are available upon request from the corresponding author.

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
