# Peer review of "Impact of Heat Stress on Milk Yield, Milk Fat-to-Protein Ratio, and Conception Rate in Thai–Holstein Dairy Cattle: A Phenotypic and Genetic Perspective"

_animals, 2024, doi:10.3390/ani14203026_

Round 1
Reviewer 1 Report
Comments and Suggestions for Authors
This article contains new and interesting topics for genetic improvement for heat tolerance of dairy cattle. However, for analysis, you have to contain effects of days in milk (DIM) or stage of lactation (SOL) in the mathematical model. In addition, recently, many articles that relevant to your research, are published that contains findings for threshold values or genetic parameters for heat tolerance. I recommend that you add those articles in the discussion section.
Details:
L38: Genetic parameters usually do not include effects but include heritability, genetic evaluation, and variances.
L59: Please explain in more detail with articles which strategies are being used in which countries. Are genetic evaluations or QTLs used in selection strategies in those countries?
L113: Did the weather data contain missing records?
L140: You have to include the effects of days in milk (DIM) for MY and FPR. And if it is possible, stage of lactation effect also have to contain in the model for CR. Because, those effects are well known as significant factors influencing their traits. Inclusion or not-inclusion of those effects might be affected estimated genetic parameters. If the models do not take them into account, you should mention them in the discussion.
Table 3: Please add heritabilities under heat conditions (for example, THI80).
Author Response
To Reviewer,
Response to Reviewer 1 Comments
This article contains new and interesting topics for genetic improvement for heat tolerance of dairy cattle. However, for analysis, you have to contain effects of days in milk (DIM) or stage of lactation (SOL) in the mathematical model. In addition, recently, many articles that relevant to your research, are published that contains findings for threshold values or genetic parameters for heat tolerance. I recommend that you add those articles in the discussion section.
We are grateful for your critical reading and efforts to improve the quality of the manuscript. We hope that the revised manuscript will meet your expectations. Our responses to each comment are listed below.
Details:
L38: Genetic parameters usually do not include effects but include heritability, genetic evaluation, and variances.
Response 1: In line 38, we are unclear about the question, but if you want us to remove the standard error values from the article, we have done so. See lines 37-38.
L59: Please explain in more detail with articles which strategies are being used in which countries. Are genetic evaluations or QTLs used in selection strategies in those countries?
Response 2: We have provided additional details, with references, on the genetic selection methods used in the selection strategy for heat stress adaptation. “Genetic selection methods such as genetic evaluation [14–17], marker-assisted selection [18,19], Genome-Wide Association Studies [20–23], and genomic selection [2,24,25] are widely used in countries like Australia, Kenya, Mexico, the United States, Tunisia, and Thailand to study the effects of heat stress on dairy cattle phenotypes and genetics. Especially, genetic evaluation is often the preferred method due to its suitability for large datasets and lower cost compared to other approaches.” See lines 65-70.
L113: Did the weather data contain missing records?
Response 3: Before analyzing the genetic parameters, we used SAS software to verify the data linkage between the weather data and all phenotypic data. The weather data were recorded digitally, ensuring no data loss.
L140: You have to include the effects of days in milk (DIM) for MY and FPR. And if it is possible, stage of lactation effect also have to contain in the model for CR. Because, those effects are well known as significant factors influencing their traits. Inclusion or not-inclusion of those effects might be affected estimated genetic parameters. If the models do not take them into account, you should mention them in the discussion.
Response 4: We have already accounted for the influence of days in milk (DIM) on MY and FPR in the genetic model by grouping DIM into 30-day intervals, referred to as months in milk. In our previous study, we found no difference in results between using months in milk or days in milk effects, but the former allowed for faster estimation of genetic parameters. See line 164.
In addition, dividing days in milk into 30-day periods, referred to as months in milk, provides a more detailed analysis of lactation stages compared to the traditional division into early, mid, and late stages. This method allows for a more precise adjustment of fixed effects on MY, milk FPR, and CR traits.
We therefore aim to use the months in milk effect as a fixed adjustment in the genetic model, as done previously.
Table 3: Please add heritabilities under heat conditions (for example, THI80).
Response 5: In Table 3, we presented the variance components and heritability values for MY, milk FPR, and CR traits under heat stress at the "threshold point of heat stress" (THI76). As your suggestion, we have included the variance components and heritability values at THI80 for comparison. See Table 3.
Best Regards
Wuttigrai Boonkum
Corresponding author

Reviewer 2 Report
Comments and Suggestions for Authors Comments to the manuscript entitled “Impact of Heat Stress on Milk Yield, Milk Fat-to-Protein Ratio, and Conception Rate in Thai–Holstein Dairy Cattle: A Phenotypic and Genetic Perspective”. The manuscript has novelty as in the changing climate scenario it is inevitable to design such studies to understand the implications of heat stress on Thai Holstein dairy cattle. However, before the manuscript is considered for publication, the following comments have to be addressed. 1. The significance of Thai cattle to the economy of farmers to be highlighted in the introduction 2. The authors must improve the discussion part establishing the correlation between phenotypic and genotypic traits with heat tolerance in Thailand cattle 3. There few lastest references missing. I would suggest the authors to update few referencesAuthor Response
To Reviewer,
Response to Reviewer 2 Comments
Comments to the manuscript entitled “Impact of Heat Stress on Milk Yield, Milk Fat-to-Protein Ratio, and Conception Rate in Thai–Holstein Dairy Cattle: A Phenotypic and Genetic Perspective”. The manuscript has novelty as in the changing climate scenario it is inevitable to design such studies to understand the implications of heat stress on Thai Holstein dairy cattle. However, before the manuscript is considered for publication, the following comments have to be addressed.
We are grateful for your critical reading and efforts to improve the quality of the manuscript. We hope that the revised manuscript will meet your expectations. Our responses to each comment are listed below.
- The significance of Thai cattle to the economy of farmers to be highlighted in the introduction.
Response 1: We have added data and references to show the importance of dairy production in Thailand as follows: “Dairy farming contributes 12.6% to Thailand's gross agricultural output, driven by increasing demand for animal-based food domestically and internationally [3,4]. Most farms are small-scale, providing income stability to rural communities, especially in the Central and Northeastern regions, which produce 70% of the country's milk [5]. The dairy industry meets domestic demand, strengthens food security, and promotes self-sufficiency. Enhancing productivity through improved breeding is crucial for ensuring future economic resilience [6].” See lines 51-58.
- The authors must improve the discussion part establishing the correlation between phenotypic and genotypic traits with heat tolerance in Thailand cattle.
Response 2: We appreciate your point about strengthening the discussion regarding the correlation between phenotypic and genotypic traits with heat tolerance in Thai cattle. We have thoroughly revised our discussion section to address this point.
We have elaborated on the negative correlations observed between milk production (MY and FPR), reproductive performance (CR), and heat tolerance and provided a more detailed explanation of the mechanisms involved. We have also highlighted the importance of considering both genetic and environmental factors when developing breeding strategies for heat tolerance in Thai cattle. Please find the revised Discussion section. See lines 381-419.
- There few latest references missing. I would suggest the authors to update few references.
Response 3: We have updated the manuscript with latest references to strengthen the revised manuscript, as detailed in the references below. See the reference section.
Latest references as follows:
- Jitmun, T.; Kuwornu, J.K.M.; Datta, A.; Anal, A.K. Farmers’ perceptions of milk-collecting centres in Thailand’s dairy industry. Pract. 2019, 29, 424–436.
- Chanchaidechachai, T.; Saatkamp, H.; Inchaisri, C.; Hogeveen, H. Analysis of epidemiological and economic impact of foot-and-mouth disease outbreaks in four district areas in Thailand. Vet. Sci. 2022, 9, 904630.
- König, S.; Chongkasikit, N.; Langholz, H.-J. Estimation of variance components for production and fertility traits in Northern Thai dairy cattle to define optimal breeding strategies. Anim. Breed. 2005, 48, 233–246.
- Chen, L.; Thorup, V.M.; Kudahl, A.B.; Østergaard, S. Effects of heat stress on feed intake, milk yield, milk composition, and feed efficiency in dairy cows: A meta-analysis. Dairy Sci. 2024, 107, 3207–3218.
- Ellett, M.D.; Rhoads, R.P.; Hanigan, M.D.; Corl, B.A.; Perez-Hernandez, G.; Parsons, C.L.M.; Baumgard, L.H.; Daniels, K.M. Relationships between gastrointestinal permeability, heat stress, and milk production in lactating dairy cows. Dairy Sci. 2024, 107, 5190–5203.
- Li, L.; Wang, Y.; Li, C.; Wang, G. Proteomic analysis to unravel the effect of heat stress on gene expression and milk synthesis in bovine mammary epithelial cells. Sci. J. 2017, 88, 2090–2099.
- Mbuthia, J.M.; Mayer, M.; Reinsch, N. Modeling heat stress effects on dairy cattle milk production in a tropical environment using test-day records and random regression models. Animal 2021, 15, 100222–
- dos Santos, M.M.; Souza-Junior, J.B.F.; Dantas, M.R.T.; Costa, L.L.M. An updated review on cattle thermoregulation: physiological responses, biophysical mechanisms, and heat stress alleviation pathways. Sci. Pollut. Res.2021, 28, 30471–30485.
- Hansen, P.J. Reproductive physiology of the heat-stressed dairy cow: implications for fertility and assisted reproduction. Proc. In: 35th Annual Meeting of the European Embryo Transfer Association (AETE); Murcia, Spain, September 12th and 14th, 2019.
- Kasimanickam, R.; Kasimanickam, V. Impact of heat stress on embryonic development during first 16 days of gestation in dairy cows. Rep.2021, 11, 14839.
- Fathoni, A.; Boonkum, W.; Chankitisakul, V.; Duangjinda, M. An appropriate genetic approach for improving reproductive traits in crossbred thai–holstein cattle under heat stress conditions. Sci. 2022, 9, 163.
- Habimana, V.; Ekine-Dzivenu, C.C.; Nguluma, A.S.; Nziku, Z.C.; Morota, G.; Chenyambuga, S.W.; Mrode, R. Genes and models for estimating genetic parameters for heat tolerance in dairy cattle. Genet. 2023, 14, 1127175.
- Soumri, N.; Carabaño, M.J.; González-Recio, O.; Bedhiaf-Romdhani, S. Modelling heat stress effects on milk production traits in Tunisian Holsteins using a random regression approach. Anim Breed. Genet. 2024, 1–15.
- Luna-Azuara, C.G.; Montaño-Bermúdez, M.; Calderón-Chagoya, R.; Ríos‑Utrera, A.; Martínez‑Velázquez, G.; Vega‑Murillo, V.E. Genetic diversity of SNPs associated with candidate genes for heat stress in Coreño Creole cattle in Mexico. Anim. Health. Prod.2024, 56, 71.
- Fang, H.; Kang, L.; Abbas, Z.; Hu, L.; Chen, Y.; Tan, X.; Wang, Y.; Xu, Q. Identification of key genes and pathways associated with thermal stress in peripheral blood mononuclear cells of Holstein dairy cattle. Genet. 2021, 12, 662080.
- Hu, L.; Fang, H.; Abbas, Z.; Luo, H.; Brito, L.F.; Wang, Y.; Xu, Q. The HSP90AA1 gene is involved in heat stress responses and its functional genetic polymorphisms are associated with heat tolerance in Holstein cows. Dairy Sci. 2024, 107, 5132–5149.
- Sigdel, A.; Abdollahi-Arpanahi, R.; Aguilar, I.; Peñagaricano, F. Whole genome mapping reveals novel genes and pathways involved in Milk production under heat stress in US Holstein cows. Front Genet. 2019, 10, 1–10.
- Cheruiyot, E.K.; Haile-Mariam, M.; Cocks, B.G.; MacLeod, L.M.; Xiang, R.; Pryce, J.E. New loci and neuronal pathways for resilience to heat stress in cattle. Rep.2021, 11, 16619.
- Czech, B.; Wang, Y.; Szyda, J. Genome-wide association study of heat stress response in Bos taurus. 2023, 1–10.
- Nguyen, T.T.T.; Bowman, P.J.; Haile-Mariam, M.; Pryce, J.E.; Hayes, B.J. Genomic selection for tolerance to heat stress in Australian dairy cattle. Dairy Sci. 2016, 99, 2849–2862.
- Gutierrez-Reinoso, M.A.; Aponte, P.M.; Garcia-Herreros, M. Genomic analysis, progress and future perspectives in dairy cattle selection: A review. Animals 2021, 11, 599.
Best Regards
Wuttigrai Boonkum
Corresponding author

Reviewer 3 Report
Comments and Suggestions for Authors
It was a pleasure reading your article and learning the research you have done. It is an important matter to address heat stress resilience in cattle in tropical and subtropical areas.
Some suggestions:
Line 78: Mentions that FPR affects cow health. It is actually cow health and/or the diet that affect FPR. I believe we need to reword this sentence.
Line 79: Conception rate needs to be defined in a sentence, as Europe, USA and Australasia may have different definitions. For instance Heat detection and heat expression of the cow play a crucial role in our herds. Pregnancy rate is the main measure we use in herds.
In example if we have 10 eligible cows and 2 are AI'd (heat detection rate of 50%), and 1 get pregnant. My Conception rate is 50% which is considered good but my pregnancy rate is only %10 which is very low!
Line 93: WHat are the Thai-Holstein crossbred cattle, crossed with? Jersey? Swiss?..
Line 100: I find the numbers not adding up. You have 69535 records. As in cow numbers? Then you have 168124 records of MY and FPR. And then you have 21278 CR records. So are these included in the 69535 records or the 168124?
Also cows with records: 21278 and Cows with pedigrees add up to 67476. Where did 69535 come from?
Needs a more detailed write-up.
Line 107: Age at first service.... So were these all first calf heifers?
Line 188: For the milk yield in the different groups, are you just talking mean production per day? Or did you use 305 day lactation average? Did you account for Days in Milk?
Line 256-258: Maybe make the sentence smaller and does not require "Moreover". Simplify the sentence. They're all showing strong negative correlations.
Line 290: Maybe say: "Heat stress negatively impacts dairy cattle performance".
In The paragraph from 293 to 312, are these studies all on Holstein cattle? It's best to differentiate if they are not.
Line 392: Severity of weather... Can mean rain, hail or something else. Why not just say: as THI increases, the rate of....
All the best
Comments on the Quality of English Language
Good use of the English language
Author Response
To Reviewer,
Response to Reviewer 3 Comments
It was a pleasure reading your article and learning the research you have done. It is an important matter to address heat stress resilience in cattle in tropical and subtropical areas.
We are grateful for your critical reading and efforts to improve the quality of the manuscript. We hope that the revised manuscript will meet your expectations. Our responses to each comment are listed below.
Some suggestions:
Line 78: Mentions that FPR affects cow health. It is actually cow health and/or the diet that affect FPR. I believe we need to reword this sentence.
Response 1: We have revised the sentence as follows: “a lack of energy in the body leads to increased lipolysis, which boosts fat synthesis in the mammary glands. Simultaneously, insufficient intake of digestible carbohydrates to meet the body's demands reduces protein synthesis by gut bacteria. This imbalance is reflected in a change in the milk fat-to-protein ratio (FPR), an indirect indicator of the cow's health and reproductive status, ultimately leading to decreased fertility and broader health issues [42,43].” See lines 93-98.
Line 79: Conception rate needs to be defined in a sentence, as Europe, USA and Australasia may have different definitions. For instance Heat detection and heat expression of the cow play a crucial role in our herds. Pregnancy rate is the main measure we use in herds. In example if we have 10 eligible cows and 2 are AI'd (heat detection rate of 50%), and 1 get pregnant. My Conception rate is 50% which is considered good but my pregnancy rate is only %10 which is very low!
Response 2: The definition of conception rate is defined. See lines 98-100.
Line 93: What are the Thai-Holstein crossbred cattle, crossed with? Jersey? Swiss?.
Response 3: We have added the sentence about the Thai-Holstein crossbred cattle as follows: “In Thailand, crossbreeding between Bos taurus breeds (such as Holstein, Jersey, Brown Swiss, and Red Dane) and Bos indicus breeds (such as Sahiwal, Red Sindhi, Brahman, and Thai Native) is widely practiced. However, the majority of crossbred dairy cattle (>95%) are the result of crosses between Holstein and either Sahiwal or Thai Native breeds [5,28].” See lines 86-90.
Line 100: I find the numbers not adding up. You have 69535 records. As in cow numbers? Then you have 168124 records of MY and FPR. And then you have 21278 CR records. So are these included in the 69535 records or the 168124?
Also cows with records: 21278 and Cows with pedigrees add up to 67476. Where did 69535 come from?
Needs a more detailed write-up.
Response 4: We sincerely appreciate your advice. The mistake was due to our own incorrect writing. To align with the information in Table 1, we have revised it as follows. “A total of 168,124 records for MY and milk FPR, along with 21,278 records for CR (conception status determined by ultrasound 30 days after breeding, where 2 = success and 1 = failure), were collected from 21,278 first-lactation crossbred Thai–Holstein cows. These records, provided by the Bureau of Biotechnology in Livestock Production, Department of Livestock Development, Thailand, were gathered between 1999 and 2017.” See lines 120-125.
Line 107: Age at first service.... So were these all first calf heifers?
Response 5: yes, all cows in our dataset were first-calf heifers.
Line 188: For the milk yield in the different groups, are you just talking mean production per day? Or did you use 305 day lactation average? Did you account for Days in Milk?
Response 6: We focused on the effects of heat stress on daily milk yield and its influence on genetic expression rather than using the cumulative 305-day yield data for several reasons. Daily data offer a more detailed and accurate reflection of how heat stress impacts cows in real-time. Heat stress can cause short-term fluctuations in milk yield, and daily data help detect these subtle changes that might be masked in the 305-day totals. Additionally, the effects of heat stress vary across the lactation period, particularly during summer months. Daily data allow us to identify specific periods when cows are most affected, unlike 305-day data, which provide an overall view without precise timing. Daily milk yield can also be directly linked to specific environmental conditions (e.g., temperature and humidity), offering a more accurate assessment of heat stress impacts. In contrast, 305-day data dilute these effects, making it harder to assess specific environmental influences. Genetic expression in response to heat stress is dynamic, and daily data enable us to analyze how genes respond to acute stress events, providing deeper insights into the genetic mechanisms involved. The use of aggregated 305-day data may overlook important genetic responses to environmental changes, limiting genetic understanding. Finally, daily data help farmers and researchers develop more targeted management strategies to mitigate heat stress, such as cooling systems or adjusted feeding regimens, which are not possible with retrospective 305-day data.
Besides, we have already accounted for the influence of days in milk (DIM) on MY and FPR in the genetic model by grouping DIM into 30-day intervals, referred to as months in milk. See line 164.
Line 256-258: Maybe make the sentence smaller and does not require "Moreover". Simplify the sentence. They're all showing strong negative correlations.
Response 7: We have revised the sentence as your suggestion as follows: “All three traits – MY, milk FPR, and CR – showed strong negative permanent environmental correlations with heat stress, with values of -0.68, -0.67, and -0.71, respectively.” See lines 274-276.
Line 290: Maybe say: "Heat stress negatively impacts dairy cattle performance".
Response 8: We have revised the sentence as your suggestion. See line 305.
In The paragraph from 293 to 312, are these studies all on Holstein cattle? It's best to differentiate if they are not.
Response 9: Our study investigated the effects of heat stress on dairy cows. In the discussion (lines 308–328), we have included data and references from research on various breeds, not limited to Holsteins. We believe that our findings are relevant and applicable to all dairy breeds worldwide. Therefore, we kindly request to retain the current discussion and references as they are.
Line 392: Severity of weather... Can mean rain, hail or something else. Why not just say: as THI increases, the rate of....
Response 10: We have revised the sentence as your suggestion. See line 423.
Best Regards
Wuttigrai Boonkum
Corresponding author

Round 2
Reviewer 1 Report
Comments and Suggestions for Authors
The manuscript has been appropriately improved.
L35: I'm sorry. I was on the wrong line, it should be L35, not L38.
In "genetic parameters (heritability values, genetic correlations, and permanent environmental effects)", "permanent environmental effects" will be removed.
There is no need to delete standard errors.